# Zeolites Enhance Soil Health, Crop Productivity and Environmental Safety

Mousumi Mondal [1], Benukar Biswas [1], Sourav Garai [1], Sukamal Sarkar [1,2], Hirak Banerjee [1], Koushik Brahmachari [1], Prasanta Kumar Bandyopadhyay [3], Sagar Maitra [4], Marian Brestic [5,6,*], Milan Skalicky [6], Peter Ondrisik [7] and Akbar Hossain [8,*]

[1] Department of Agronomy, Bidhan Chandra Krishi Viswavidyalaya (BCKV),
    Nadia 741252, West Bengal, India; mou.mousumi98@gmail.com (M.M.); kripahi@yahoo.com (B.B.);
    garai.sourav93@gmail.com (S.G.); sukamalsarkarc@yahoo.com (S.S.); hirak.bckv@gmail.com (H.B.);
    brahmacharis@gmail.com (K.B.)
[2] Office of the Assistant Director of Agriculture, Bhagwangola-II Block, Directorate of Agriculture,
    Government of West Bengal, Murshidabad 742135, West Bengal, India
[3] Department of Agricultural Chemistry and Soil Science, Bidhan Chandra Krishi Viswavidyalaya (BCKV),
    Nadia 741252, West Bengal, India; pkb_bckv@rediffmail.com
[4] Department of Agronomy, Centurion University of Technology and Management,
    Paralakhemundi 761211, Odisha, India; sagar.maitra@cutm.ac.in
[5] Department of Plant Physiology, Slovak University of Agriculture, Nitra, Tr. A. Hlinku 2,
    949 01 Nitra, Slovakia
[6] Department of Botany and Plant Physiology, Faculty of Agrobiology, Food and Natural Resources,
    Czech University of Life Sciences Prague, Kamycka 129, 165 00 Prague, Czech Republic; skalicky@af.czu.cz
[7] Department of Environment and Zoology, Slovak University of Agriculture, Nitra, Tr. A. Hlinku 2,
    949 01 Nitra, Slovakia; peter.ondrisik@uniag.sk
[8] Bangladesh Wheat and Maize Research Institute, Dinajpur 5200, Bangladesh
[*] Correspondence: marian.brestic@uniag.sk (M.B.); akbarhossainwrc@gmail.com (A.H.)

**Abstract:** In modern days, rapid urbanisation, climatic abnormalities, water scarcity and quality degradation vis-à-vis the increasing demand for food to feed the growing population necessitate a more efficient agriculture production system. In this context, farming with zeolites, hydrated naturally occurring aluminosilicates found in sedimentary rocks, which are ubiquitous and environment friendly, has attracted attention in the recent past owing to multidisciplinary benefits accrued from them in agricultural activities. The use of these minerals as soil ameliorants facilitates the improvement of soil's physical and chemical properties as well as alleviates heavy metal toxicity. Additionally, natural and surface-modified zeolites have selectivity for major essential nutrients, including ammonium ($NH_4^+$), phosphate ($PO_4^{2-}$), nitrate ($NO_3^-$), potassium ($K^+$) and sulphate ($SO_4^{2-}$), in their unique porous structure that reduces nutrient leaching. The slow-release nature of zeolites is also beneficial to avail nutrients optimally throughout crop growth. These unique characteristics of zeolites improve the fertilizer and water use efficiency and, subsequently, diminish environmental pollution by reducing nitrate leaching and the emissions of nitrous oxides and ammonia. The aforesaid characteristics significantly improve the growth, productivity and quality of versatile crops, along with maximising resource use efficiency. This literature review highlights the findings of previous studies as well as the prospects of zeolite application for achieving sustenance in agriculture without negotiating the output.

**Keywords:** soil amelioration; resource use efficiency; water conservation; nutrient retention; heavy metal toxicity

## 1. Introduction

The increasing pressure of the population leads to a higher food demand, and at least 50% more food production is required to meet the demand of people by 2050, without any

scope of horizontal land diversification [1,2]. Therefore, intensive agricultural practices in food and nutritional security force the use of irrational chemical inputs, water and heavy machinery. More than two-thirds of the renewable water resources are exclusively used by agricultural activities, resulting in uneven water sharing with the other sectors [2–4]. Furthermore, the consequences of intensive practices are the degradation of soil and water qualities, such as depletion of soil organic carbon and inherent soil nutrient status, heavy metal contamination and residual fertilizer and/or pesticide mixing with groundwater vis-à-vis surface water resources, that dwindle crop productivity and ultimately the per capita food grain availability [5]. Long-term intensive farming activities make the agricultural land unproductive, resulting in low soil retention capacity. The most important element, nitrogen, is widely used in agricultural systems, although its use efficiency in nitrogenous fertilizers rarely exceeds 50% as it is mostly lost through denitrification, leaching and volatilisation [6]. Moreover, irrational application of nitrogenous fertilizers facilitates easy $NO_3^+$ discharge from soil to groundwater, causing negative anthropogenic impacts on the groundwater quality and public health hazards such as methemoglobinemia, cancer of digestive organs, eutrophication in water bodies and production of greenhouse gases such as nitrous oxide ($N_2O$) through the denitrification process [7–10]. Phosphate ($PO_4^{3+}$) is another major nutrient in fertilizer, also responsible for eutrophication in water bodies [11]. Therefore, soil nutrient retention is a major concern in modern agriculture to account for maximum nutrient use efficiency, improve the soil nutrient status and prevent groundwater contamination [12–14]. Nutrient use efficiency and better plant growth are highly related to soil's physical and chemical properties. In this context, the application of soil amendments, more particularly natural or organic amendments, has great importance for the long-term reclamation of soil's physicochemical properties [15–17]. Zeolites are naturally occurring, alkaline-hydrated aluminosilicates with more than 50 different forms [18,19] and a wide range of applications such as soil-binding agents and nutrient supplements for animal and aquatic lives. Additionally, they can be used as heat storage materials and solar refrigerators, both absorber and adsorber; ion-exchanging elements; molecular sieving agents; and catalysing agents in various chemical reactions [20,21]. In agriculture, the importance of zeolites has been realised to a greater extent with their varying applicability (Figure 1) [20]. Natural zeolites are being considered as good soil ameliorating substances, having good water and nutrient holding capacity (WHC); it improves infiltration rate, saturated hydraulic conductivity, cation exchange capacity, and prevents water losses from deep percolation [22–26]. Moreover, zeolites could be used as fertilizer and chelating agent [27]. Zeolites minimize the rate of nutrient release from both organic and inorganic fertilizers and enable better nutrient availability throughout the crop growth stages [27]. The improvement of the wide range of agronomic and horticultural crops in respect to growth, yield and quality traits with the application of zeolites has been well reported by various researchers [28–33]. Additionally, zeolite can effectively absorb heavy metals such as cadmium (Cd), lead (Pb), nickel (Ni), anions like chromate ($CrO_4^{2-}$) and arsenate ($AsO_4^{-3}$), and organic pollutants such as volatile organic compounds (VOCs) including benzene, toluene, ethylbenzene, and xylene (BTEX) from soil or water body [34–36]. Acknowledging all the aforesaid advantages, the applications of zeolites in the agricultural research field have been widely gained importance since the last two decades (Figure 2), evidenced by the chronological ascending trend of the publication rate accessed from "Scopus" online database with the keywords of "zeolit", "soil remediation", "water retention", "nutrient retention", "crop production" and "heavy metal toxicity". Several earlier findings reported the applicability of zeolites on soil properties along with water and nutrient retention capacity, crop yield and heavy metal toxicity. Therefore, it is high time to give importance to zeolites application in agricultural activities and this review article gives a comprehensive assessment on the sources of zeolites, their structure and properties, and wide application in agriculture with the special consideration of soil properties, resource conservation, pest management, pollution control and crop productivity.

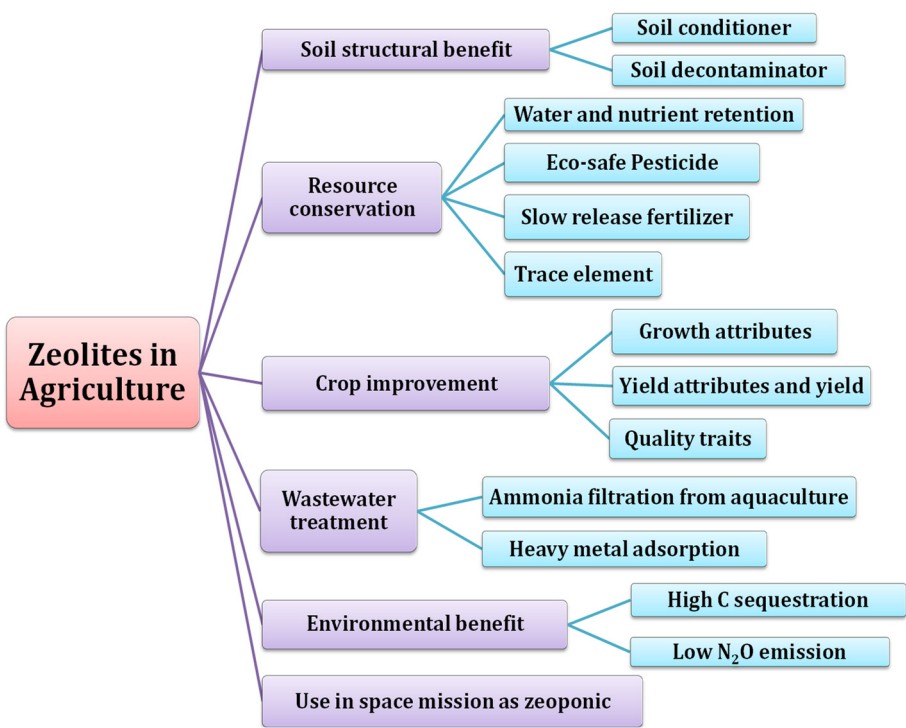

**Figure 1.** Multidimensional Uses of Zeolites in Agriculture.

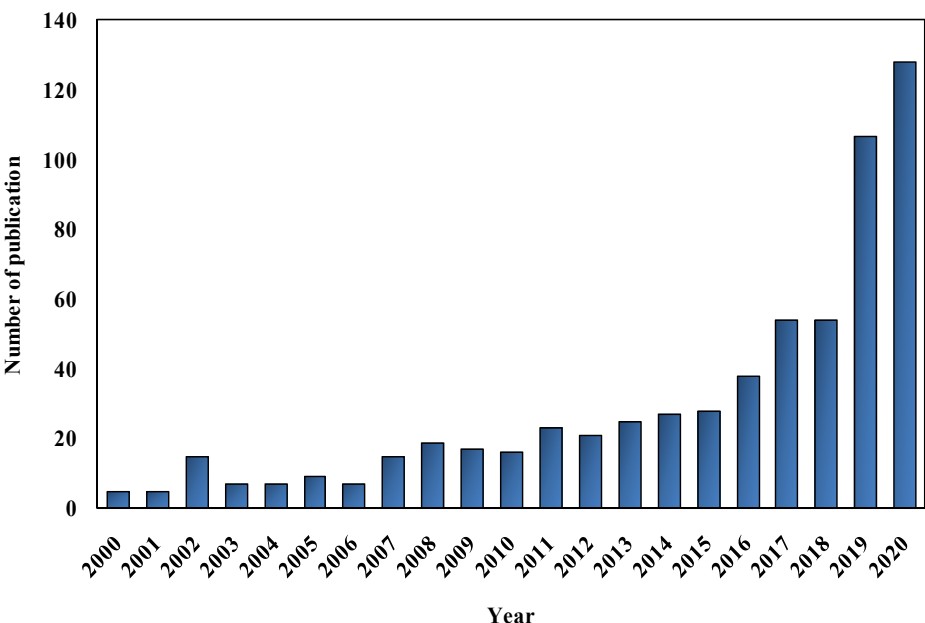

**Figure 2.** The Trend of Annual Publications on Zeolite Applications in Agriculture for the Last Two Decades. Source: Scopus Preview [37].

## 2. Origin, Structure and Properties of Zeolites

The word zeolites refer to 'boiling stones' because of their ability to froth when heated to about 200 °C. The first time, the mineral zeolites are identified by a Swedish mineralogist Alex Fredrik Cronstedt in 1756 [38]. However, zeolites production was started commercially in the 1960s [38]. China contributes ~75% market share of total zeolites production, followed by Korea (8%), the United States (3%), and Turkey (2%) [39]. In India, the maximum zeolitic enriched soil is found in the state of Maharashtra followed by Karnataka, Gujrat, Andhra Pradesh and West Bengal (Figure 3).

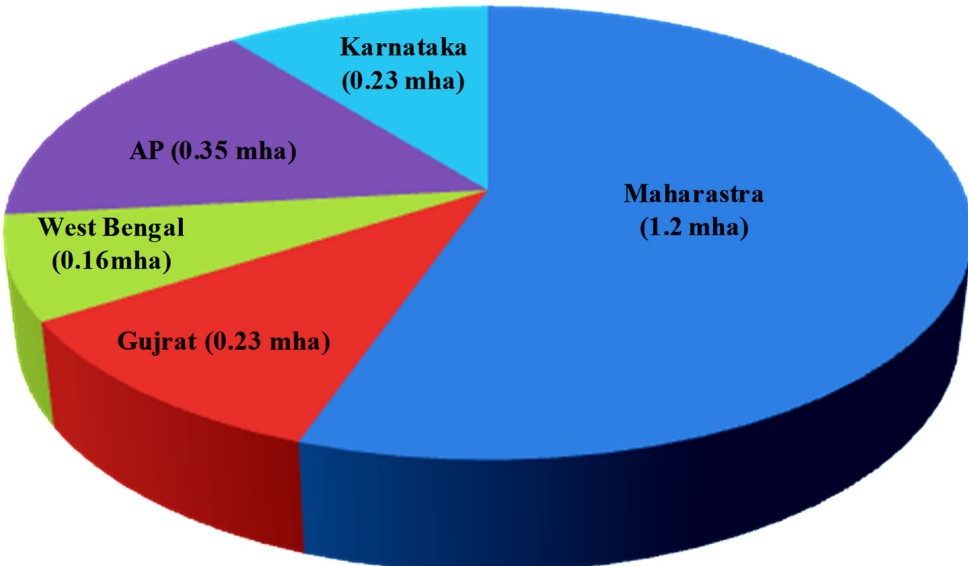

**Figure 3.** Distribution of Zeolitic Soil in India. Modified from Bhattacharyya et al. [40].

Structurally, zeolite is comprised of aluminosilicate ($AlO_4$ and $SiO_4$) tetrahedrons, joined into three-dimensional frameworks and seems like a honeycomb structure (Figure 4) [41]. The cages in the porous structure of zeolite are approximately 12 Å in diameter, interlinked through the channels of 8 Å diameter, includes 12 tetrahedrons rings [42]. Depending on the minerals, the pores are interlinked to form long wide channels which facilitate easy molecular movement into and out of the zeolite structure. The negative charge of aluminum ions in the zeolite structure is balanced by positively charged cations.

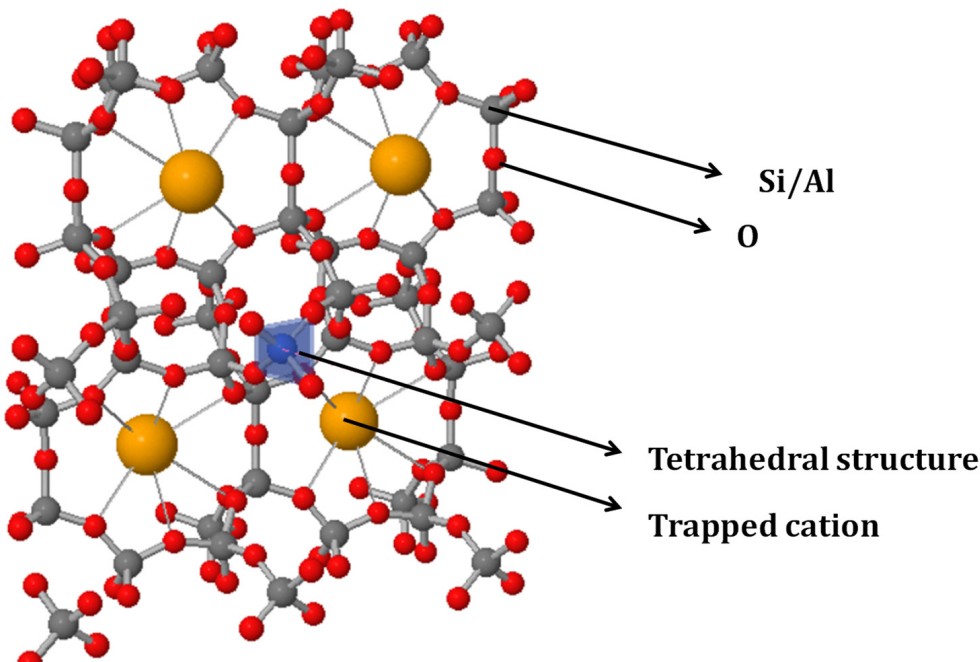

**Figure 4.** The Tetrahedral Framework of Clinoptilolite Zeolite. Modified from IZA [43].

The general empirical formula that refers to a zeolite structure is $M_{2n}O \cdot Al_2O_3 \cdot xSiO_2 \cdot yH_2O$. M refers to any alkali or alkaline earth cation; the valence of the cation is indicated by n, x ranges between 2 and 10, and y ranges between 2 and 7, with structural cations comprising $Si^{2+}$, $Al^{3+}$ and $Fe^{3+}$, and exchangeable cations $K^+$, $Na^+$ and $Ca^{2+}$ [44]. The

spacious porous structure with large channels in zeolite structure makes it unique in nature as compared to other silicate minerals [45]. Natural zeolites are loaded with aforesaid cations with various considerable properties such as higher cation exchange capacity (CEC) than normal soil, ranges between 100 and 200 centimol (+) kg$^{-1}$ [46], free water storage within their structural channels, and also have a great ability of ion adsorption in large surface area. Zeolites can adsorb or exchange various cations viz. strontium (Sr) and cesium (Cs); heavy metals like zinc (Zn), cadmium (Cd), lead (Pb), manganese (Mn), nickel (Ni), chrome (Cr), iron (Fe), and copper (Cu) [34]; anions such as chromate ($CrO_4^{2+}$) and arsenate ($AsO_4^{3+}$) [35]; and numerous organic pollutants mentioned earlier [36]. Other useful physical and chemical properties of zeolites include high void volume (~50%), low density (2.1–2.2 g/cm$^3$), excellent molecular sieving properties and high cation selectivity exclusively for ammonium, potassium, and cesium ions [40]. Physical characteristics of some naturally occurring zeolites are summarized in Table 1. In respect to pore diameter Zeolites have been classified by Flanigen [47], viz. (i) Small-pore (0.3–0.45 nm diameter with 8 rings), (ii) Medium-pore (0.45–0.6 nm diameter with 10 rings), (iii) Large-pore (0.6–0.8 nm diameter with 12 rings), and (iv) Extra-large pore zeolites (0.8–1.0 nm diameter with 14 rings).

**Table 1.** Physical Characteristics of Some Naturally Occurring Zeolites.

| Zeolites | Porosity (%) | Channel Dimensions (Å) | Heat Stability | Ion Exchange Capacity (meq g$^{-1}$) | Specific Gravity (g cm$^{-3}$) | Bulk Density (g cm$^{-3}$) | References |
|---|---|---|---|---|---|---|---|
| Analcine $Na_{10}(Al_{16}Si_{32}O_{96})\cdot16H_2O$ | 18 | 2.6 | High | 4.55 | 2.24–2.29 | 1.85 | Sangeetha and Baskar [42] |
| Chabezite$(Na_2Ca)_6$ $(Al_{12}Si_{24}O_{72})\cdot40H_2O$ | 47 | 3.7 × 4.2 | High | 3.85 | 2.50–2.10 | 1.45 | IZA [43] |
| Clinoptilolite $(Na_3K_3)(A_{16}Si_{30}O_{72})\cdot24H_2O$ | 34 | 3.9 × 5.4 | High | 2.17 | 2.15–2.25 | 1.15 | IZA [43] |
| Erionite $(AlCaH_{60}KNaO_{36}Si_2{}^{+3})$ | 35 | - | High | 3.12 | 2.02–2.08 | 1.51 | Hemingway and Robie [44] |
| Heulandite $(Ca_4)(Al_8Si_{28}O_{72})\cdot24H_2O$ | 39 | 4.0 × 5.5 | Low | 2.90 | 2.18–2.20 | 1.69 | Sangeetha and Baskar [42] |
| Mordenite$(Na_8)(Al_8Si_{40}O_{96})\cdot24H_2O$ | 28 | 2.9 × 5.7 | High | 4.30 | 2.12–2.15 | 1.70 | Chmielewska and Lensỳ [45] |
| Philipsite$(NaK)_5(Al_5Si_{11}O_{32})\cdot20H_2O$ | 31 | 4.2 × 4.4 | Moderate | 3.32 | 2.15–2.20 | 1.58 | Chmielewska and Lensỳ [45] |
| Faujasite $(Na_{58})(Al_{58}Si_{134}O_{384})\cdot240H_2O$ | 47 | 7.4 | High | 3.38 | - | - | Hemingway and Robie [44] |
| Laumonitte$(Ca_4)(Al_8Si_{16}O_{48})\cdot16H_2O$ | 34 | 4.6 × 6.3 | Low | 4.25 | - | - | Sangeetha and Baskar [42] |
| Linde A$(Na_{12})(Al_{12}Si_{12}O_{48})\cdot27H_2O$ | 47 | 4.4 | High | 5.47 | - | - | Sangeetha and Baskar [42] |
| Linde X$(Na_{86})(Al_{86}Si_{106}O_{384})\cdot264H_2O$ | 50 | 7.4 | High | 4.72 | - | - | Sangeetha and Baskar [42] |

## 3. Impacts of Zeolite Application in Agriculture

### 3.1. Improvement of Soil Physical Properties

Soil physical properties include bulk density, particle density, aeration, soil porosity, water holding capacity in which bulk density is the basic soil property that influences the total porosity and topsoil stability [48]. The application of zeolites in light texture soil reduces the bulk density that modifies the water holding capacity and soil air porosity [49]. However, total porosity is not influenced significantly [49]. In a previous study, Xiubin and Zhanbin [3] opined the natural zeolite mainly mordenite with less than 0.25 mm size to the fine-grained calcareous loess which had low WHC. Result revealed that after 25 h of water addition to treated and normal soils, the zeolites applied soil resulted in higher

water content (Figure 5). They also reported that water holding capacity in zeolites treated soil increased 0.4–1.8% in drought condition while 5–15% in normal situation as compared to non-treated soil.

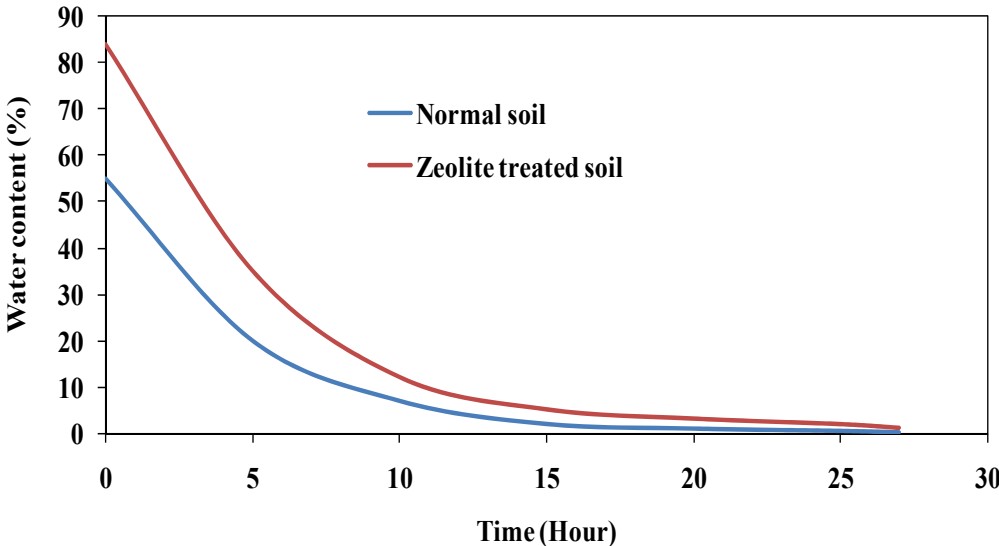

**Figure 5.** Soil Water Content as Influenced by Zeolitic Soil. Modified from Xiubin and Zhanbin [3].

In another study, the effect of modified $Ca^{+2}$ type zeolite on sand dune soil was determined where irrigated was given with saline water. Sand dune soil samples were treated with the three different rates of zeolite i.e., 5 kg m$^{-2}$, 1 kg m$^{-2}$ and no zeolite (control) and irrigated with seawater diluted to electrical conductivity (EC) levels of 3 and 16 deciSiemens per metre (dSm$^{-1}$). Results showed that soil with 5 kg zeolites m$^{-2}$ enhanced soil water as well as salt content, accounting for 20 and 1.4% higher than no application of zeolite [16]. The concentration of cations namely $Ca^{2+}$, $K^+$, $Na^+$, $Mg^{2+}$ is increased with the increasing soil salinity. The findings were attributed to the fact that zeolite increases the cation exchange capacity, and subsequent cations holding on the surface soil, and release them at the expense of salts in the saline water [22]. Thus, the low salt accumulation in subsurface soil facilitates low salt stress on plants and creates a better environment for plant growth. Lowering of particle size with the application of zeolites in sandy soil might be another reason for higher water holding capacity. Higher pore volumes in zeolites facilitate greater water holding in their structures [49]. Such structures are not damaged by water particles during surface evaporation and/or reabsorption. Zeolites may be considered as the permanent water reservoir. Retention of soil moisture in longer duration, particularly during dry periods helps to mitigate drought-induced abiotic stresses and enable plants to withstand in dry spell; zeolites also facilitate to rapid rewetting and the lateral water spreading throughout the root zone during the time of irrigation that reduces the timing of water application [41]. Soil amelioration with zeolites increases the water availability to plants by 50% [42]. Application of zeolite @ 10 g kg$^{-1}$ soil could maintain maximum water percentage (8.4%) at field capacity and delay in permanent wilting point in sandy loam soils [50]. Al–Busaidi et al. [16] reported that the existence of fine particles and micropores in zeolites slowed down the deep percolation of soil water. The infiltration rate is inversely proportional to zeolites application (Figure 6) indicating the higher soil water residence and subsequent restriction in nutrient and salt leaching. Xiubin and Zhanbin [3] observed that the mixing of zeolites with fine grain calcareous loess soil increased the infiltration rate by 7–30% and 50% in a gentle and steep slope respectively. Furthermore, run-off and subsequent soil erosion were reduced with the zeolites application and the sedimentation also found to be decreased by 85% and 50% in a gentle and steep slope respectively. Interestingly, a combination of zeolites and selenium application check the water deficit oxidative damages in plants [51]. Colombani et al. [52]

quantified the changes in flow and transport parameters induced by the addition of zeolites in a silty-clay soil and reported that $NH_4^+$ enriched zeolites enhanced the capacity of water retention in silty-clay soil, thus diminishing the water and solute losses. Maximum irrigation water productivity (0.81 kg m$^{-3}$) under limited irrigation supply was registered with the supplemental application of zeolites (21% ww$^{-1}$) along with urea, while the minimum water productivity (0.48 kg m$^{-3}$) was observed under full irrigation supply and exclusive urea application [53]. Bernardi et al. [54] also observed that concentrated zeolites as a sand-soil amendment increase at least 10% of soil-water retention and 15% of available water to plants. Zeolite increases the periods between the starting of rainfall and runoff occurrence. Rainfall intensity with 10 mm per hr. results in the beginning of runoff within 15 min in normal soil while in zeolites (20%) treated soil runoff starts after 30 min of rainfall occurrence [55].

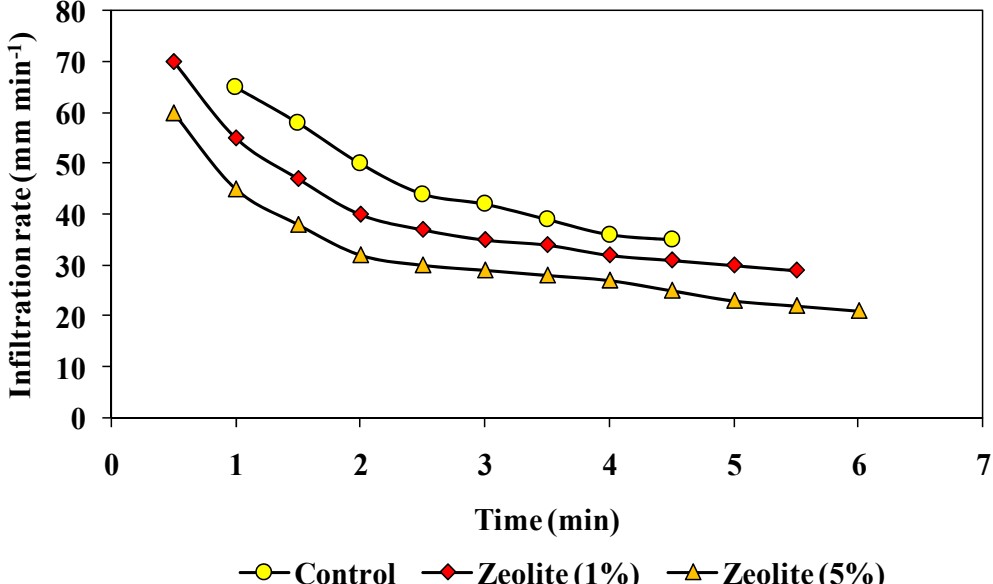

**Figure 6.** Soil Infiltration Rate as Influenced by Different Rates of Zeolite Application. Modified from Al-Busaidi et al. [16].

Zeolites help to improve the water-stable aggregates in soil. As per example, nano zeolite with 30% concentration increased the mean weight diameter of water-stable aggregates by 0.735 mm [56]. With the use of this property, Moritani et al. [57] reported that the incorporation of 10% artificial zeolites in sodic soils resulted in improved wet aggregate stability ranged between 22.4% and 59.4% depends on the soil textural classes. Cario et al. [58] categorized the soil with average assessment ranking 'good' and 'excellent' in terms of water-stable aggregates and degree of soil aggregation in Vertisols and they showed the application of zeolite along with chemical fertilizers or organic manure (Zeolite @ 7.5 t h$^{-1}$ + sugarcane filter cake @ 22.5 t h$^{-1}$) improved the soil properties from good to excellent. Sepaskhah and Yousefi [59] conducted an experiment to justify the effect of various rate of calcium-potassium zeolite on the pore velocity of water in the soil they observed higher pore water velocity (35 and 74%) with the application of 4 and 8 g zeolite kg$^{-1}$ soil respectively. Changes of soil physical properties with the Zeolite application in thin (heavy) textured medium-thin textured, and medium coarse (light) textured soil was observed by Gholizadeh-Sarabi and Sepaskhah [60] reported that in fine and medium texture soil, zeolites application at the rate of 4 and 8 g kg$^{-1}$ of soil at the low salinity level (0.5 and 1.5 dS m$^{-1}$) and 16 g zeolites kg$^{-1}$ soil at the high salinity level (3.0 and 5.0 dS m$^{-1}$) increased saturated hydraulic conductivity significantly while in coarse texture soil similar rate of zeolites application reduced the saturated hydraulic conductivity considerably. They also assumed that zeolites application in the heavy (clay loam) and medium-textured

soil (loam) changed the shape and size of the soil pores and resulted in an improvement of soil structure and the water movement in these soils. Zeolites application alleviates the adverse effect of salinity on hydraulic conductivity and thus it would prevent waterlogging in heavy and medium soil textures. In case of sandy soils, zeolites addition would be appropriate to decrease the hydraulic conductivity and the transferability of water that results in low deep percolation and loss of soil water. However, Razmi and Sepaskhah [61] reported that the application of zeolite ($8\ \mathrm{g\ kg^{-1}}$) in silty clay soils significantly improved the hydraulic conductivity. They also established that the soil treated with zeolite resulted in 50% less crack depth in dry puddled soil with pre-application of zeolites in comparison to no zeolites application. A similar observation was also recorded after the first and second irrigation in puddled condition (Figure 7).

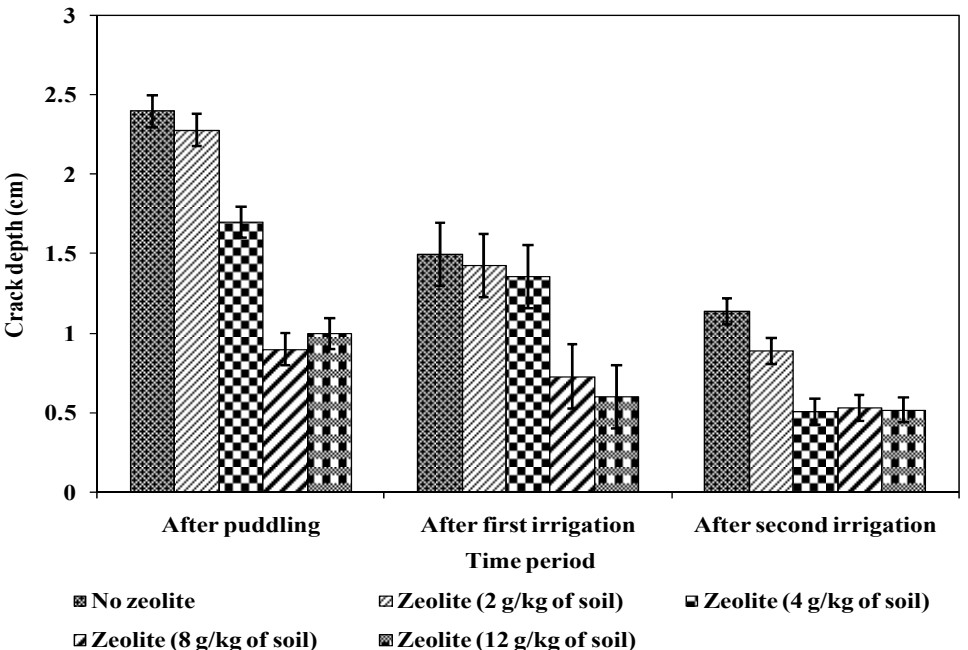

**Figure 7.** Effect of Zeolite on Crack Depth in Puddled Transplanted Rice. Modified from Razmi and Sepaskhah [61].

Furthermore, the sorptivity of clay-loam soil was reduced with a higher rate of zeolites application as reported by Gholizadeh-Sarabi and Sepaskhah [60]. However, a contrasting result was observed in the case of sandy-loam and loamy soil. Proper use of water is the immediate need in agriculture to ensure food security with available water resources; hence, technologies that enhance water use efficiency are being widespread. The aforesaid discussions indicate that zeolites addition positively influence the inter-particle porosity as well as total porosity, bulk density, hydraulic conductivity, infiltration rate, and cation exchange capacity of soil that ultimately accelerates the soil water content. Additionally, the open pore network channels into zeolites structure mainly play the significant roles' in water retention. The summarization of zeolitic impacts to the wide range of soils in Table 2 indicates that the use of zeolites as a soil ameliorant would be a welcome strategy in agriculture.

**Table 2.** Physical Properties of Soils as Influenced by Zeolites Application.

| Types of Zeolite | Application Rate (ww$^{-1}$) | Soil Textural Classes | Changes in Soil Physical Properties | | | References |
|---|---|---|---|---|---|---|
| | | | Water Content | Infiltration Rate | Hydraulic Conductivity | |
| Clinoptilolite | 1–15% | Clay, loamy sand, sand | • 20% increase at 10% zeolite application rate in sandy soil | • Infiltration rate reduction with a higher rate of application | • Decreased in sandy and loamy soils; Increased in clay soil | Mahabadi et al. [15] |
| Mordenite | – | Calcareous loess | • 0.4–1.8% and 5–15% increase in drought conditions and normal conditions | • 7–30% increase with gentle slopes >50% increase with steep slope | – | Xiubin and Zhanbin [3] |
| Non-specified natural zeolite | 0.4%, 0.8%, 1.6% and no zeolitie | Sandy loam | – | • Highest sorptivity at 0.4% (0.5 dS m$^{-1}$ salinity) <br> • Lowest sorptivity at 1.6% | • Decreased at 0.8 and 1.6% | Gholizadeh-Sarabi and Sepaskhah [60] |
| | | Loam | – | • Highest sorptivity at 1.6% (all salinity levels) | • Maximized at 1.6% and 0.4% at 3–5 dS m$^{-1}$ and 0.5–1.5 salinity respectively | |
| | | Clay loam | – | • Lowest sorptivity at 1.6% and 0.8% at 0.5–3 and 5 dS m$^{-1}$ salinity | • Maximized at 1.6% and 0.8% at 3–5 and 0.5–1.5 dS m$^{-1}$ salinity level | |
| Synthetic zeolite (Ca$^{2+}$-type) | 1% and 5% | Sand dune soil | • Increased at 5% level | • Reduction in infiltration rate (more for higher application rate) | | Al-Busaidi et al. [16] |
| Stilbite | 3.33, 6.67, and 10% | Sandy soil | • 10%, 38% and 67% increase with 3.33%, 6.67% and 10% level respectively | – | | Bernardi et al. [54] |
| Clinolite and Ecolite | 15.85% | – | – | – | • Increase (More in Ecolite over Clinolite) | Githinji et al. [62] |

### 3.2. Nutrient Retention

Zeolites positively influence the physical, chemical, and biological properties of soil directly or indirectly which in turns improves the nutrient dynamics as well as nutrient retention capacity. Zeolitic minerals have high CEC which attributes to high $NH_4^+$ sorption selectivity as a consequence of the electrostatic attraction between positively charged $NH_4^+$ and negatively charged sites in zeolite structure [63,64]. The effective diffusion coefficient was around $4–5 \times 10–12$ $m^2$ $s^{-1}$ for ammonium and sodium ions respectively in clinoptilolite [65,66]. The adsorption capacity of zeolites for these ions is determined by isotherms and kinetics and this adsorption property is used for various purposes such as wastewater treatment, heavy metal removal. Clinoptilolite generally exhibits a high selectivity for $NH_4^+$ ion, having theoretical CEC of 2.16 cmol (+) $kg^{-1}$ [67]. I on adsorption efficiency of zeolites are mainly depends of the factors like mass, particle size, initial concentration of cations of model solution, contact time, temperature and pH [68,69]. Additionally, modification of zeolites surface with strong acids accelerates the cation sorption capacity [70]. The modification of natural zeolites includes pretreatment by grinding and sieving, mixing with sodium salt and finally, calcinations makes a change in the pore size and surface area of zeolites, and thereby the ammonium ion uptake is increased [71].Soil application of zeolites in combination with chemical fertilizers reduces nitrogen leaching [72–74] and volatilization [75–77] slows down the mineralization process and subsequent reduction in greenhouse gases (GHGs) emission [78], and retards the nutrients release into soil solution [79,80]. In the incubation studies, researchers had clearly seen the difference in the ammonia loss with chemical fertilizers and chemical fertilizers with zeolite and reported low ammonia losses when fertilizer applied with zeolite [81,82]. Omar et al. [83] proved the significant improvement in soil exchangeable ammonium retention by 40–50% in zeolite treated soil. The leaching reduction of $NH_4^+$ and $NO_3^-$ from different nitrogenous fertilizer with the application of zeolite is depicted in Table 3. The zeourea and nano-zeourea contain 18.5% and 28% of N respectively and capable to release N up to 34 and 48 days, respectively, while from conventional urea the N releases within 4 days after application [84]. The reason behind this may be the urease activity is significantly reduced by zeolite application that lowers the nutrient release from fertilizer [85]. The slow-released nature of fertilizer helps to release their nutrient contents gradually and to coincide with the nutrient requirement of a plant [86].

**Table 3.** Leaching Reduction Percent of $NH_4^+$ and $NO_3^-$ from Different Nitrogenous Fertilizer with the Application of Zeolite.

| Soil Type | Zeolite Application Rate | Source of N | N Dose (kg $h^{-1}$) | Leaching Reduction | | References |
|---|---|---|---|---|---|---|
| | | | | $NH_4^+$ | $NO_3^-$ | |
| Sand-based putting green | 10% | Ammonium Sulphate | 293 | 99% | 86% | Huang and Petrovic [87] |
| Sandy soil | 0.8% | Ammonium Sulphate | 32 | >90% | – | Zwingmann et al. [88] |
| Loamy sand | 5% | Ammonium Sulphate | 200 [†] | 83% | – | Mackown and Tucker [89] |
| Sandy loam | 9 [*] | Urea | 270 | – | 36% | Golamhoseini et al. [90] |
| Silty loam | 4% | Wastewater | 14.2 [‡] | – | 54.9% | Taheri-Sodejani et al. [91] |

[*] With the unit of t $ha^{-1}$; [†] With the unit of mg $kg^{-1}$; [‡] With the unit of mg $L^{-1}$.

Urea saturated zeolite chips have also been developed elsewhere. Piñón-Villarreal et al. [92] experimented to assess the leaching loss from urea ammonium nitrate solution (UAN32) where 443 mg total N was present per liter of solution. They observed that 82% reduction in leaching loss happened from the pure clinoptilolite zeolite loaded column in comparison to the column of loamy sand. In a sorption experiment, Piñón-Villarreal et al. [92] reported more than 90% $NH_4^+$ absorption by zeolite incorporated soil in initial several minutes. Very small particle size with a greater surface area of zeolitic minerals accelerates the stabilization of exchange equilibrium in only a few hours. Zeolite minerals also protect the

conversion of $NH_4^+$ to $NO_3^+$ through the nitrification process. The latter is more prone to leach out into the soil and facilitates to groundwater contaminations [59]. The small pores in zeolite crystal lattice structure (4–5 Å) in which cations like ammonium can easily adsorb, do not give access to the nitrifying microorganisms into the pores [93]; thus, nitrification does not take place easily in zeolites treated soil. One of the most usefulness of zeolite is utilized in compost making a way to convert agricultural farm waste into valuable organic amendments. However, a significant amount of N losses take place during the time of composting [94]. In an experiment, Ramesh and Islam [95] confirmed that the application of 14–21% zeolite in fresh manure resulted in low ammonium loss. Zeolite also could absorb volatile substances such as acetic acid, butanoic acid, skatole and isovaleric acid and also could effectively control the odor released during composting [96,97].

The extent of reduction in total nitrogen and even phosphorus losses with the application of zeolite into organic manure was successfully reported by Murnane et al. [98]. The reason behind the low N losses from manure is the high specific selectivity of zeolites to ammonium ($NH_4^+$) that helps in holding this ion during volatilization. Moreover, the existences of small internal channels protect $NH_4^+$ from rapid nitrification by microbes [99]. Interestingly, zeolites not only help to protect the N loss but also reduces P leaching; however, it helps in reducing $NO_3^-$ leaching greater than P leaching [53,100,101]. Being alkaline in nature and the presence of negative charges, zeolite ameliorated soil improves soil P availability through lowering of soil acidity, soil exchangeable Al, and Fe [101–103]. These help in less P fixation by metal oxyhydroxides. Moreover, zeolites supplementation triggers more P uptake by enhancing the exchange- induced dissolution mechanisms as follows [102]:

$$RP(rock\ phosphate) + NH_4^+ + zeolite \rightarrow Ca - zeolite + NH_4^+ + H_2PO_4^- \qquad (1)$$

In this reaction, released Ca is adsorbed on the zeolite surface due to high CEC and as a result, more rock phosphate will be dissolved with lowering $Ca^{2+}$ activity in the solution. This system releases the $NH_4^+$ and $PO_4^{3-}$ ions. The addition of clinoptilolite zeolites with a 75% recommended rate of fertilizers showed comparable total and available P with the existing recommended dose without any zeolite application [31]. In this experiment, the addition of clinoptilolite zeolites also helped to reduce Al as well as soil acidity that resulted in low P fixation to soil colloid. A similar trend of observation was recorded by Zheng et al. [104], accounted for 14.1% higher available P with the application of zeolite relative to non-zeolite treatment. Antoniadis et al. [105] also reported an increase in P recovery efficiency of 4.02% due to zeolite application in acidic soil as compared to no zeolite application. The slow-release nature of zeolite in P release was observed by Bansiwal et al. [106] resulted in the continuous phosphate release even after 1080 h of continuous percolation from zeolite loaded modified phosphorous surface, while within only 264 h phosphate from potassium dihydrogen phosphate ($KH_2PO_4$) was exhausted.

Rather than N and P zeolites have strong selectivity on $K^+$ than $Na^+$, $Ca^{2+}$, and $Mg^{2+}$ that makes it difficult to remove $K^+$ from exchange sites, facilitating greater absorption of $K^+$ by plant root hairs through the ion exchange within root and zeolite [107]. The losses of $K^+$ by surface runoff and groundwater leaching can be reduced by supplementing the zeolites as slow-release fertilizer [108]. For example, the application of zeolites in municipal compost to investigate the $K^+$ release pattern resulted in six times less leaching loss from the zeolitic compost as compared to normal compost [109]. Additionally, Williams and Nelson [110] observed that in a soil-less medium $K^+$ saturated clinoptilolite recorded 23% less leaching of $K^+$ over-controlled control substrate. Moraetis et al. [109] reported that there was 18-fold increase in bioavailable K when zeolites were added through kinetic experiment to the soil-compost mixture, suggesting high potassium affinity in the soil-compost-zeolite mixture. Zeolite is considered as nano-enhanced green application as it adsorbs molecules at relatively low pressure [111,112]. Zeolite coated fertilizers have higher potential in water absorption and retention, and this coating materials retard the nutrient release rate from soil applied fertilizers, especially in sandy and sandy loam soil [113].

Similar nutrient retention ability of zeolites in secondary nutrients such as S was registered by Li and Zhang [114] who revealed that after leaching with 50 pore volumes, 85% of the pre-loaded $SO_4^{2+}$ remained on the zeolite modified S fertilizer. Moreover, the initial $SO_4^{2+}$ concentration in the leachate of S-loaded surfaced modified zeolite was found to be lowered, in comparison with the non-zeolitic sulfur sources. In addition to clinoptilolite, nano-zeolite based S fertilizer is also comprised of epistilbite zeolite. The findings from an experiment conducted by Thirunavukkarasu and Subramanian [115] exhibited that $SO_4^{2+}$ was available even after 912 h of continuous percolation from S loaded modified nano-zeolite, while $SO_4^{2+}$ from $(NH_4)_2SO_4$ was depleted within 384 h. The presence of a huge number of channels, pores, and cages in the structure of the zeolite which helps in holding the $SO_4^{2+}$ tightly might be the reason behind the slow release of this secondary nutrient from surface modified nano-zeolite [115].

The increase in micronutrient use efficiency with zeolites supplementation was also registered in previous literatures [33,116–118]. Sheta et al. [116] reported the ability of five natural zeolites and bentonite minerals to adsorb and release of zinc and iron as natural zeolites have a greater affinity to these micronutrients. Iskander et al. [117] found 74.7% and 84.63% are readily extractable by DTPA (diethylene-triamine pentaacetic acid) extractant (0.005 M DTPA + 0.01 M CaCI$_2$ + 0.1 M triethanolamine, adjusted to pH 7.30) after three successive extractions of Zn and Mn, respectively and rest were retained by zeolite. Yuvaraj and Subramaniannano [119] reported that nano-zeolite adsorbed more Zn and the adsorption rate obtained with the nano-zeolite appeared to be efficient adsorbents for Zn. They also observed that ZnSO$_4$ released the Zn up to 200 hours whereas micronutrients from nano-zeolite were releasing even after 800 h (Figure 8). The better availability of micronutrients in soil with zeolite application ultimately facilitates to greater micronutrients contents in plants. Ozbahce et al. [33] resulted in significantly higher Zn, Mn and Cu content in bean leaves with the maximization of zeolite application up to 90 kg ha$^{-1}$ (Figure 9). From the above-mentioned discussions, it can be concluded that the zeolite application accelerates the availability of primary, secondary and micronutrients in soils and subsequent plant uptake (Figure 10), and its application is most significant in arid and semi-arid regions that suffer from high water and nutrient scarcity all-time.

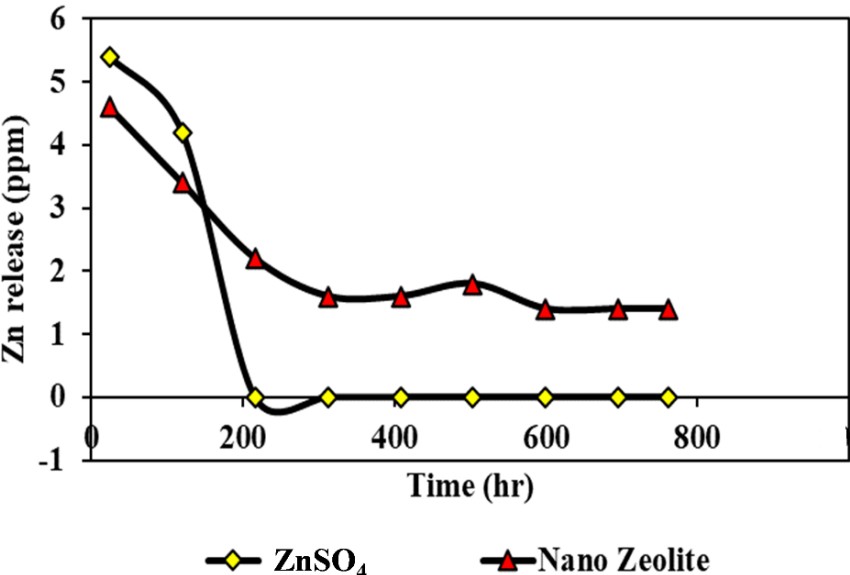

**Figure 8.** Zn Release Pattern with Time Duration as Influenced by Nano Zeolite Application. Modified from Yuvaraj and Subramaniannano [119].

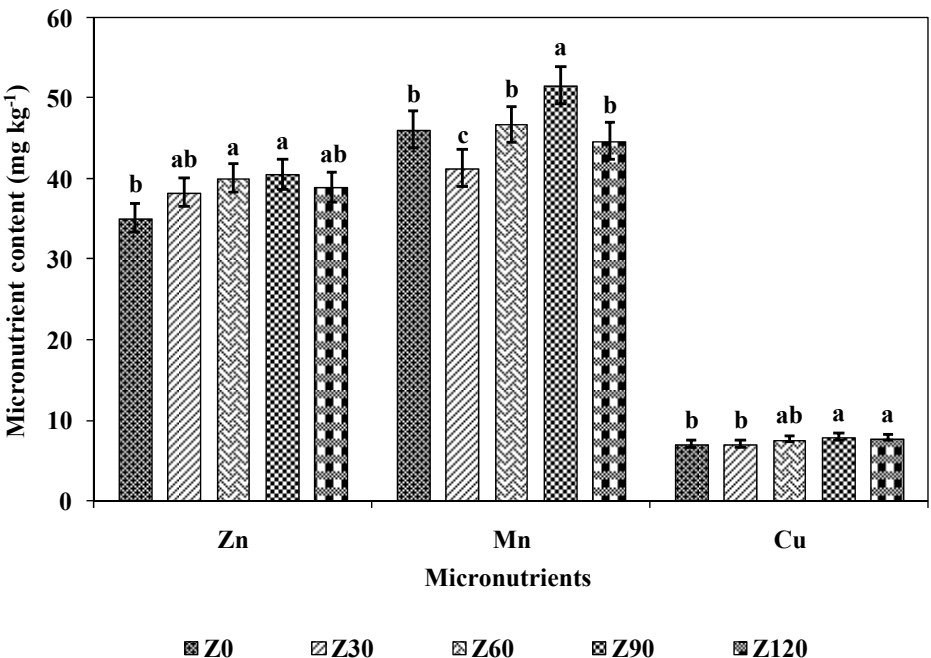

**Figure 9.** Micronutrient content in bean leaves with different levels of zeolite application ($Z_0$: 0; $Z_{30}$: 30; $Z_{60}$: 60; $Z_{90}$: 90; $Z_{120}$: 120 t ha$^{-1}$); within treatments, different letters indicate significant differences at $p \leq 0.05$(otherwise statistically at par); error bars represent the least significant difference value. Modified after Ozbahce et al. [33].

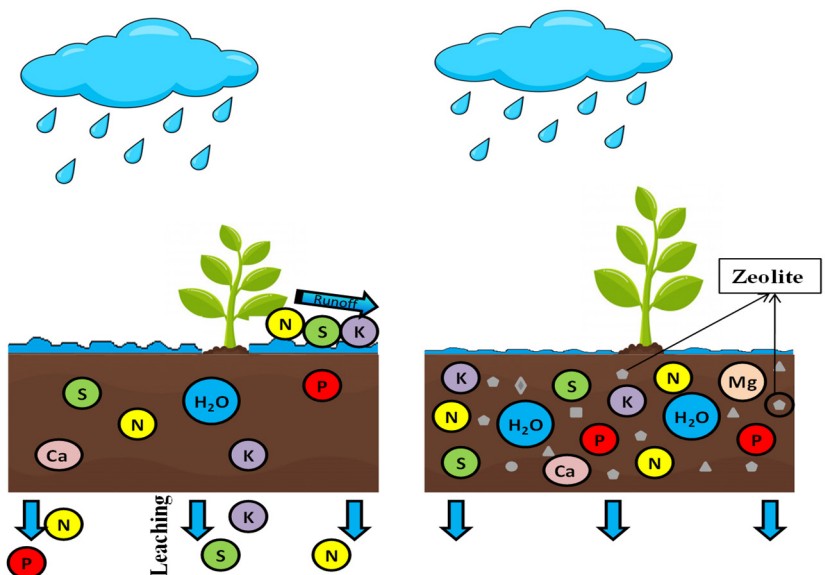

**Figure 10.** Effectiveness of Zeolite on Water and Nutrient Retention in Soil. Modified from Nakhli et al. (2014).

### 3.3. Environmental Impact

The addition of zeolites increases the C sequestration and subsequent soil C stock as compared to untreated soil [120,121]. According to Aminiyan et al. [56], application of zeolite (30%) along with crop residues (5%) to wheat could maintain the highest amount of organic carbon in light and heavy fractions. Soil organic matter even in the light fraction is highly correlated with N mineralization and subsequent soil management practices. The light fraction of soil organic matter (SOM) is not only sensitive to changes in management practices but also correlates well with the rate of N mineralization. Periodical measurements of $N_2O$ and $N_2$ emissions in fields from the applied cow urine or potassium nitrate ($KNO_3$)

each at 200 kg N ha$^{-1}$ with and without the addition of zeolite (clinoptilonite) showed that zeolite significantly lowered the total N$_2$O emissions by 11% from urine treated soils.

Specific channel size enables zeolite to act as molecular gas sieves. Wang et al. [122] recommended the use of zeolite as an amendment to reduce GHGs emission from duck manure as they found almost 27% of GHG emissions reduction from zeolite treated soil than no zeolite application. Additionally, low NO$_3^-$ and PO$_4^{2-}$ leaching from zeolite amended soil helps to prevent groundwater pollution as well as surface water contamination and subsequent eutrophication [59]. They claimed that the better retention of anions in zeolite structure might be the reason for less leaching loss. Zeolite prevents rapid mineralization by preventing the entry of nitrifying bacteria into its structure and thus reduces the emission of N$_2$O [99].

### 3.4. Slow Release of Herbicides

Being porous in nature along with a well-ordered structure, zeolites are considered as potential substances for storage and release of organic guest molecules. The most hydrophobic solid form of zeolite 'ZSM 5' adsorbs triazine group of herbicides in the compartmentalized intra-crystalline void space and release them slowly [123]. Furthermore, ZSM-5 was found to be restricted to the mobility of post-emergence herbicide such as paraquat [124,125]. Humic acid zeolites act as a sorbent of the herbicides belongs to the phenylurea group [126]. Clinoptilolitic turf has the potential to remove atrazine from soil and water [127,128]. Application of 2, 4–D herbicide along with zeolites results in a gradual temporal release pattern and keeps the active ingredient of herbicide in upper 0–5 cm of soil layer [129,130]. This slow-release nature of herbicide when used with zeolites improves the herbicide efficiency to control the weed floras and the prolonged effect of herbicide keeps the weed-free crop field throughout the entire crop weed competition period. Zeolite-rich nanocapsule is used as an herbicide carrier, adsorbent and retaining agent [130]. A longer retention period of zeolite added herbicide on weed leaves helps in maximizing the efficacy of the herbicidal mode of action. Interestingly, the synergistic effect between zeolite-loaded catalysts with isoproturon accelerates the visible light absorption and moreover better adsorption of recalcitrant molecules by the porous structure of zeolites [131].

### 3.5. Remediation of Contaminated Soil

Heavy metals induced soil pollution is one of the major concerns in modern agriculture. The anthropogenic activities of human, rapid industrialization and injudicious use of fertilizers without proper precaution make the soil toxic with heavy metal contamination. The solubility of heavy metal in soil is depending on complex chemical degradation and numerous factors. Among them, low soil pH is one of the major determining factors. In an acidic environment, oxides of iron, aluminum and manganese are slowly solubilized, and the primary and secondary minerals release the heavy metal into soil [132]. Soil sorption capacity is another determining factor for the retention of heavy metal ions. The ongoing concern in relation to the purity of the soil and the need to restore its original properties forced us to seek new and alternative ways of soil cleansing. Zeolite additions increase the soil pH significantly which facilitates to the heavy metal adsorption on its surface; thus, the solubility and bioavailability of heavy metals are ultimately reduced [133]. Chen et al. [134] observed that the cadmium and lead accumulation in wheat is significantly reduced with soil application of zeolite in soil. Moreover, it has been well reported that the clinoptilolite zeolite effectively controlled the heavy metal solubility including cadmium and lead up to 72% and 81% respectively [135,136]. However, this area of research needs extensive studies to find out heavy metal-specific appropriate dose and methods of zeolite application [85].

### 3.6. Wastewater Treatment

Industrial development with fast urbanization produces large quantities of wastewater that contains heavy metals, oils and organics that badly affect the aqueous environment [137]. Various efficient techniques such as solvent extraction, ion exchange and

adsorption are often used to remove those contaminants. Among them, the use of zeolites as adsorbents is most popular due to low-cost involvement, eco-friendly and poses good selectivity for toxic cations [138]. It also prevents the generation of new waste materials [139]. Furthermore, zeolites more specifically clinoptilolite could adsorb dyes, humic acid, phenols and phenol derivatives from the water body [140–142]. The clinoptilolite is mostly effective against metallic cations such as $Al^{3+}$, $Cd^{2+}$, $Cu^{2+}$, $Ni^{2+}$, $Pb^{2+}$, and $Zn^{2+}$ from copper mine wastewater [143]. The selectivity by clinoptilolite for heavy metals following the order: $Pb^{2+}> Cd^{2+}> Cu^{2+}> Co^{2+}> Cr^{3+}> Zn^{2+}> Ni^{2+}$ [144]. The most advantage of clinoptilolite use in wastewater treatment is it can adsorb the heavy metals at a wide range of temperature (25–60 °C), pH (1–4) and different agitation speed (0, 100, 200, 400 rpm) [145]. The greater surface area along with high cation exchange capacity makes zeolite as a good adsorbent of cations [142]. The ability of heavy metals uptake by clinoptilolite zeolite was investigated by Baker et al. 2009 and opined the high selectivity of zeolite for the discharge of $Pb^{2+}$ (98%), followed by $Cr^{3+}$, $Cu^{2+}$ and $Cd^{2+}$ with 96% selectivity within 90 min. Morkou et al. (2015) [146] reported that wastewater nutrients can be recycled and used for microalgal and cyanobacterial biomass production by using zeolite as a medium.

### 3.7. Crop Management Practices

Zeolites have been used in a wide range of field crops production such as rice (*Oryza sativa* L.), corn (*Zea mays* L.), wheat (*Triticum aestivum* L.), potato (*Solanum tuberosum* L.), soybeans (*Glycine max* L.), and other upland crops in all types of soil to improve their productivity, water, and nutrient use efficiency (NUE), also maintaining the soil ecology and environment [83,147,148]. In an experiment, Chen et al. [29] estimated the effect of different rates of zeolite in combination with different N levels on transplanted rice and concluded that the highest yield was achieved consistently when rice plant was treated with a maximum dose of N (157.50 kg ha$^{-1}$) along with zeolite supplementation (15 t ha$^{-1}$), accounting 14.90% higher than the exclusive application of N. They also revealed that yield attributing characters namely effective tillers per plant, number of grains per panicle, grain filling percentage, and 1000-grain weight were positively influenced by the higher dose of N; however, zeolite consistently increased the number of effective tillers (Figure 11). A possible explanation of these results is the slow-release characteristics of zeolite amendment that makes the essential plant nutrients available throughout the crop growth within 0–30 cm soil depth. Furthermore, the supplementary application of zeolite significantly influenced the quality traits like protein content and tasting score of rice but did not influence the head rice recovery and chalkiness of rice grain [29].

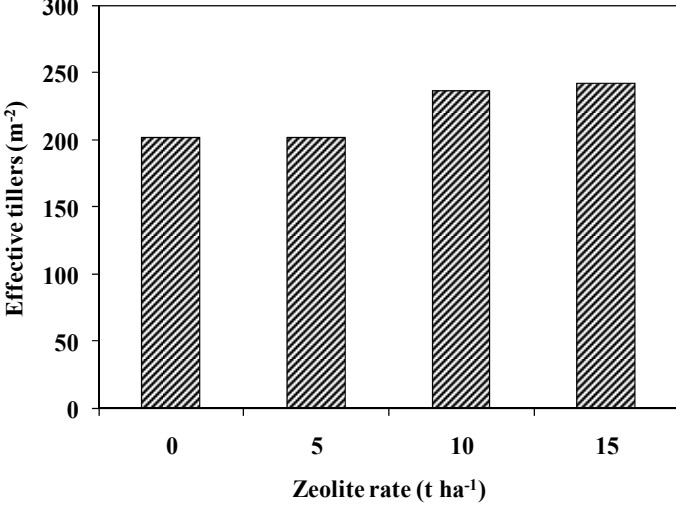

**Figure 11.** Effect of Zeolite Application on Tillering Pattern of Rice. Modified from Chen et al. [29].

In another experiment, Zheng et al. [32] evaluated the effect of zeolite application on rice under limited water condition and they confirmed that the zeolite treatment (15 t ha$^{-1}$) improved the LAI, transpiration rate and stomatal conductance (Figure 12). They observed that chalky rice rate and chalkiness were decreased by 29.6% and 41.2% respectively in zeolite treated plants as compared to the non-zeolite control. There was no significant difference in zeolite application on the starch viscosity properties. As rice quality is thought to be determined both genetically and environmentally, any improvements with zeolite application may result from better nitrogen and water availability to plants. The better crop performance and N partitioning in different parts of the rice plant with higher levels of zeolite application were depicted by Wu et al. [149]. Kavoosi et al. [150] resulted in both rice grain and straw yield increment with the application of zeolite at a certain level and thereafter decreased (Figure 13).

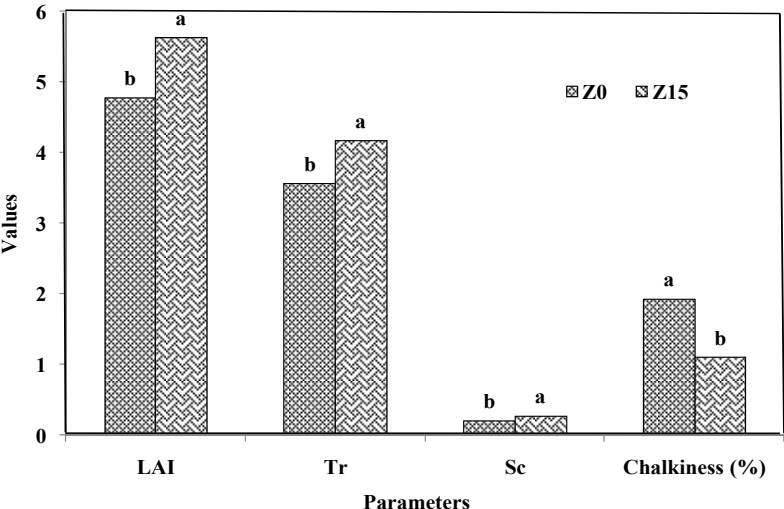

**Figure 12.** Effect of zeolite application on LAI, Tr, Sc and Chalkiness of rice. Tr: Transpiration rate (mmol m$^{-2}$ s$^{-1}$); Sc: Stomatal conductance (mol m$^{-2}$ s$^{-1}$); Z$_0$: No Zeolite; Z$_{15}$: Zeolite at 15 t ha$^{-1}$. Within treatments, different letters indicate significant differences at $p \leq 0.05$(otherwise statistically at par). Modified from Zheng et al. [32].

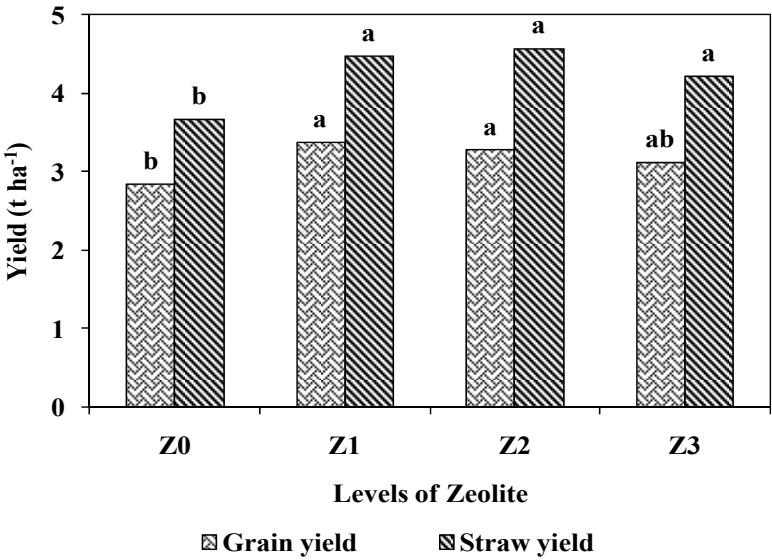

**Figure 13.** Effect of zeolite application on grain and straw yield of rice (Z$_0$: 0; Z$_8$: 8; Z$_{16}$: 16; Z$_{24}$: 24 t ha$^{-1}$). Within treatments, different letters indicate significant differences at $p \leq 0.05$(otherwise statistically at par). Modified from Kavoosi [150].

According to Wu et al. [151], the zeolites amendment significantly improved the root characteristics in terms of root length, dry weight, root diameter and volume, total root surface area, root bleeding intensity in rice plant over no zeolite application. Developed root traits may enhance nutrient transportation from the root to the above-ground parts and result in higher biomass and grain yield [152]. In previous studies, researchers confirmed that additional zeolites supply maximized the leaf area index (LAI) as well as leaf SPAD values and photosynthetic efficiency in rice plant, which might be attributed to its better ammonium retention capacity and slow-release nature that increase the better N availability to plants [53,72]. In a lowland rice production system, Sepaskhah and Barzegar [153] established the positive correlation between zeolites application and N retention in the upper soil profile. This higher availability favours better N uptake and subsequently higher nitrogen use efficiency (Figure 14). Zeolites induced rice cultivation resulted in greater apparent N recovery (65%) while 40% recovery was observed in exclusive N fertilization [150,154].

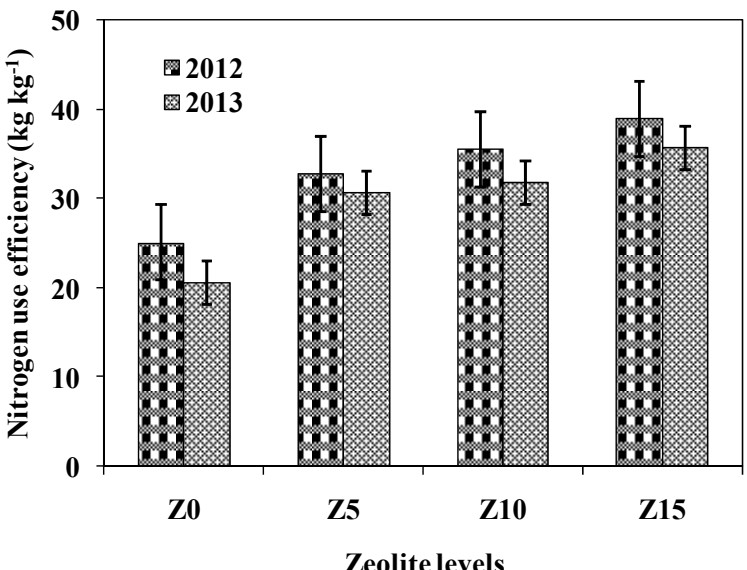

**Figure 14.** Nitrogen use Efficiency of Rice with Different Rates of Zeolite Application ($Z_0$: 0; $Z_5$: 5; $Z_{10}$: 10; $Z_{15}$: 15 t ha$^{-1}$). Modified from Chen et al. [29].

Zheng et al. [104] evaluated the consequence of zeolite and phosphorus applications in rice under different irrigation regimes and resulted in 15.2% higher water use efficiency (WUE) as well as greater leaf and stem P concentration by 20.3% and 32.7% respectively than no-zeolite control. The better water use efficiency may be attributed to higher soil water retention in the porous structure of zeolites and thus better water availability to plant [3,53]. Additionally, restriction in deep percolation and leaching beyond the crop root zone in zeolite loaded soil are major reasons for better water use efficiency [23,155].

The application of zeolite in maize cultivation was reported by Malekian et al. (2011) [156] who opined that maize plants resulted in better response to zeolite when used as a fertilizer carrier at the rate of 60 g kg$^{-1}$ of soil. The application of clinoptilolite zeolite (CZ) with a 75% recommended dose of fertilizer resulted in significantly similar cobs yield in maize as compared to the full recommended dose of fertilizer [31]. A similar trend of observation was recorded regarding dry matter production and nutrient uptake, especially N and K uptake. It is possible due to the higher cation exchange capacity and affinity of CZ to $NH_4^+$ and $K^+$ ions. More specifically, reduced nitrification, prevention of leaching and volatilization by inhibiting ureolytic activity of microorganisms in the presence of CZ facilitate better nutrients availability [157]. Moreover, the cation selectivity of the CZ in the order to $K^+ > NH_4^+ > Na^+ > Ca^{2+} > Mg^{2+}$ supports to the aforesaid observation [31,158]. Increased nitrogen-use efficiency with the application of zeolites and ensured good reten-

tion of soil-exchangeable cations, available P and $NO_3^-$ within the soil have been found by Rabai et al. [159] in maize cultivation. Low fertilizer requirement with zeolites application not only gives a similar yield but also reduces the environmental pollution in respect to nitrous oxide emission, with maintaining the economic viability. Andronikashvilf et al. [147] also suggested that the zeolite application facilitates a reduction in the recommended dose of fertilizer by 25% and maintains a positive effect for 2–3 years in upland crops production systems.

In high saline condition zeolite amendment in soil responded well in Barley crop and it was reported that zeolite at 5% level produced taller plants; accumulated maximum plant biomass and more grain yield over 1% and no zeolite application [16]. Similarly, in alkaline condition soil application of zeolites for French bean (*Phaseolus vulgaris* L.) cultivation maximized the nutrient accumulation in plant tissues. Additionally, better crop performance as well as greater water use efficiency, water productivity and crop yield were recorded from the zeolites treated plots [154]. Usually, the higher $Na^+$ content in alkaline and saline soils disturbs the soil nutritional balance and osmotic regulations in plant tissues. Zeolite provides additional $Ca^{2+}$ cations in the soil to reduce the $Na^+/Ca^{2+}$ ratio. The provision of $Ca^{2+}$ from zeolite in the growing media would alleviate the toxic $Na^+$ ions accumulation and helps in the improvement of soil structure by aggregating the soil particles [16].

Not only in cereals and pulses zeolites have significant importance in oilseed crops. An additional supply of 10-ton zeolites $ha^{-1}$ with recommended fertilizer significantly increased the seed and oil yield in safflower, accounted for 2.7 and 9.38 t $ha^{-1}$ respectively [160]. Zahedi et al. [161] evaluated the effects of zeolite and selenium applications on some agronomic traits of three Canola cultivars under drought stress. They opined that stem diameter significantly decreased due to water stress, while the application of zeolite along with selenium improved stem diameter may be attributed to better water and nutrients availability from zeolites induced soil. They reported that 10 t zeolite $ha^{-1}$ significantly improved the growth, yield attributes, and yield (Figure 15). They also observed reduced N leaching along with higher water holding capacity and CEC in alkaline soil when supplemented with 10 t zeolite as compared to normal soil. The oil yield and oil qualities such as palmitic acid, Oleic acid, Linoleic acid, Linolenic acid and Erucic acid of canola significantly improved with zeolite application (15 t $ha^{-1}$) rather than no zeolite use [28].

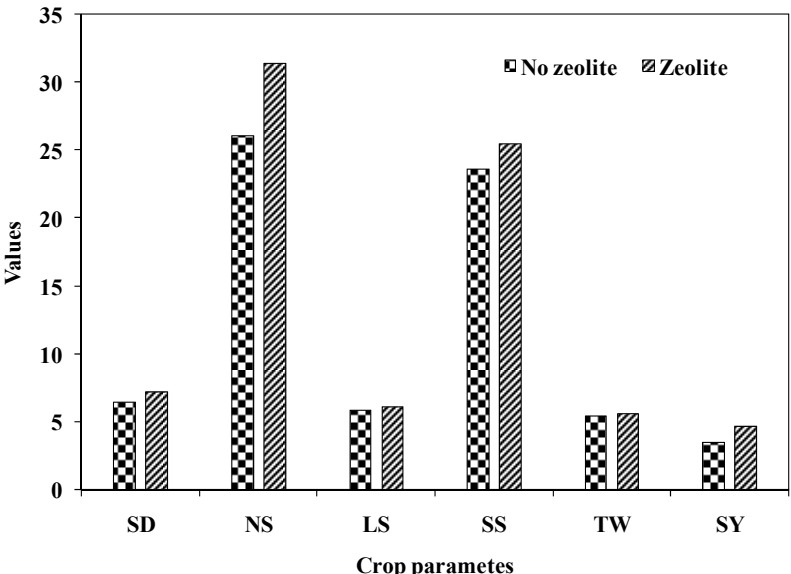

**Figure 15.** Effects of zeolite on some agronomic traits in canola. SD—Stem diameter (mm); NS—Number of siliquae; LS—Length of siliqua (cm); SS—Seeds per siliqua; TW—Test weight (g); SY—Seed yield (t $ha^{-1}$). Adapted and modified from Zahedi et al. [162].

From an experiment, the findings recorded by Ozbahce et al. [162] revealed that application of 60 t zeolite ha$^{-1}$ along with proper irrigation and nutrient management, potato yielded (39.1 t ha$^{-1}$) maximum tubers (Figure 16). They also recorded superior crop performance even under limited water supply when treated with zeolite while non-zeolite traditional practices sharply decreased the tuber yield. The interaction of zeolites and irrigation regimes was found to be significant for tuber weight, tuber diameter and crude protein percentage.

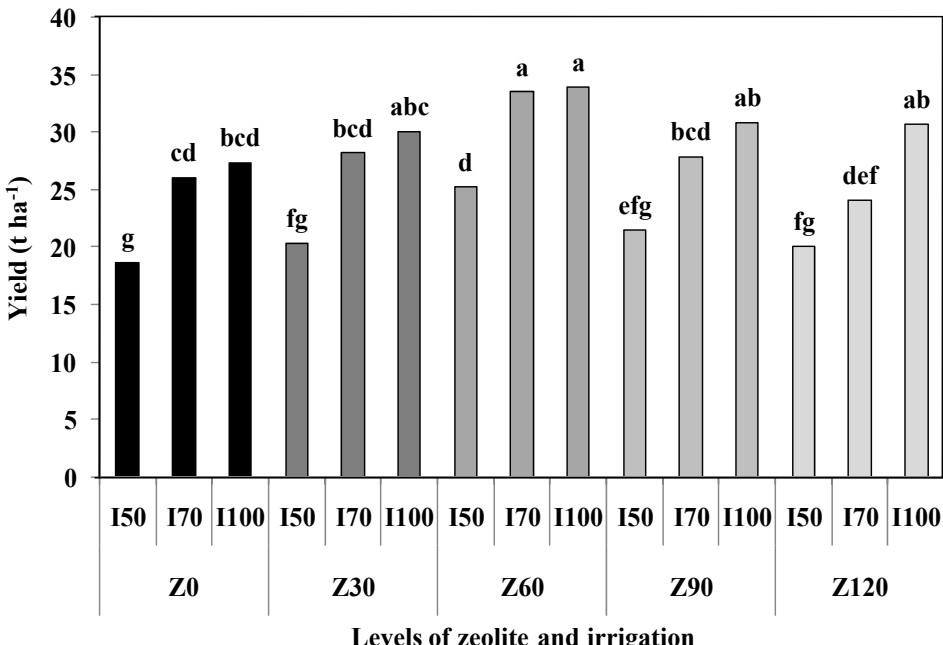

**Figure 16.** Effect of Zeolite Rates Eighth Different Levels of Irrigation on Tuber Yield of Potato (Z$_0$: 0; Z$_{30}$: 30; Z$_{60}$: 60; Z$_{90}$: 90; Z$_{120}$: 120 t ha$^{-1}$) Within treatments, different letters indicate significant differences at $p \leq 0.05$(otherwise statistically at par). (Adapted and modified from Ozbahce et al. [163]).

The effectiveness of zeolite on Peppermint (*Mentha piperita* L.) cultivation was reported by Ghanbari and Ariafar [30]. They opined that zeolite treatment significantly improved the fresh and dry leaf weight of mint and the highest value of fresh dry leaf weight was observed in 2.5 g zeolite application per kg of soil even under—water scare situation. They also observed that drought intensity was decreased with increasing the zeolite application. In 30% field capacity, zeolite application maximized the leaf dry weight from 18.54 to 32.76 g and fresh leaf weight from 41.7 to 67.14 g. Interestingly, zeolite helps to keep the essential components of mint oil such as menthol, menthone, methyl acetal, menthofuran and palegone [163,164]. Actually, these essential components are adversely affected by drought and salinity stress whereas, zeolites consist of alkali and alkaline materials and crystalline alumino—silicate which act as a water reservoir in their internal surface area during drought situation [165,166].

Numerous scientific reports were also concluded that significant positive influence on cocoa fruiting [167], eye numbers in potato tubers [160], pod and siliqua number in pulses and oilseeds [33,168], and overall development of soybean, sweet potato, wheat, bean and safflower with the application of soil—applied zeolites [160–171]. The use of Clinoptilolite—rich tuff as soil conditioner was found to be effective to improve the productivity of wheat, eggplant, carrots, and apples by 13–15%, 19–55%, 13–38% and 63% respectively [172]. Not only in field crops or vegetables, zeolite induced soil significantly improved the production as well as qualitative traits of mycelium mushroom [173]. The treatment with 30% zeolite + 70% urea resulted in a positive effect on the microbiological community in spring barley, soybean and maize [174]. Andronikashvili et al. [175] opined that the introduction of

clinoptilolite containing tuffs into soils enhanced the soil microbial population viz. bacteria, fungi and actinomycetes.

Another interesting dimension of zeolite application was introduced by the National Aeronautics and Space Administration (NASA), which developed a special type of clinoptilolite loaded plant growth media including synthetic apatite, dolomite, and several essential trace nutrients mainly for vegetable production (10% higher than non—zeolite application) in space missions, known as 'zeoponic' [176]. Life support system for regenerating and recycling the air, water and food are essentially required for the long duration Mars mission and only the growing of plants could be fulfilled this aim. The ultimate objective of zeoponic research is to develop a solid substrate that can supply all essential macro and micronutrients slowly for a long duration in a space habitat. In an experiment, Gruener et al. [176] resulted in higher biomass accumulation, root and leaf development and nutrient uptake by radish when cultivated in zeoponic as compared to normal soil. Rodriguez—Fuentes et al. [177] reported that root architecture, plant growth and yield of different vegetables, spices and strawberries, were significantly improved by zeoponic substrates without further fertilization. Researchers confirmed that the native clinoptilolite in zeoponic acts as a good source of N and K as the clinoptilolite cations are exchanged for $NH_4^+$ and $K^+$ ions [102]. Additionally, apatite and dolomite dissolution supplies $Ca^{2+}$ into soil solution. This $Ca^{2+}$ rich solution removes the $NH_4^+$ and $K^+$ ions from zeolite exchange complex and makes them more available to plants [176]. Sometimes, nitrifying bacteria are supplemented to zeoponic substrates prior to plant growth to augment the nitrification process [178]. Since most zeolites are advantageous in the growth and development of crops, however, erionite (one type of zeolite) was found to be detrimental to the proper growth of plants [179]. Therefore, the selection of an appropriate form of zeolites should be taken into consideration.

### 3.8. Used as a Pesticide

Zeolites that contain silica gel and alumina silicate crystals have been successfully tested against some stored grain pests such as lesser grain borers (Rhyzopertha dominica), rice weevils (Sitophilus oryzae), and saw—toothed grain beetles (Oryzaephilus surinamensis) [180]. Natural zeolites application at the rate of 50 g $kg^{-1}$ of maize grain were also found to be effective against maize weevil (*Sitophilus zeamais*) in accordance with Haryadi et al. [181]. Clinoptilolite was successfully investigated on organic oilseed rape fields against the pollen beetle (*Meligethes* sp.). Daniel et al. [182] observed that under dry and sunny weather condition, pollen beetles were significantly reduced by 50 to 80% with zeolite application while in rainy weather zeolite did not perform against pollen beetles. Zeolites loaded organophosphorus compound was used with success against the *Aedes aegyptii* [183]. Clinoptilolite is gaining importance as possible sorbents because it acts as a slow—release carrier and retard water contamination [184]. Clinoptilolite riched metalaxyl application on turfgrass against *Phythium* sp. resulted that the active ingredient of fungicide was prevented from groundwater contamination by clinoptilolite zeolite [185]. Actually, the adsorption of pesticide molecules is happened due to polar chemical bonds with the external surface of the microporous zeolitic minerals [108]. Additionally, the dusting of natural zeolites has been successfully tested to control the aphid population in fruit orchard [186]. Moreover, in herbicide application, pest control and in nano—sensing for pest detection, the nano—porous zeolites have been implicated as nano—capsules [187–189]. Stadler et al. [190] examined the insecticidal effect of nanostructured zeolites on two stored—grain insect species, *S. oryzae* and *R. dominica*, and found 80–100% mortality rate within 14 days after application to wheat grain. In this regard, natural zeolites may provide a cheap and reliable alternative to commercial insecticides in pest management. The insecticidal efficacy of natural zeolite on different stored grain pests is summarized in Table 4. Additional research is needed to investigate the mode of action, non—target toxicity, and the potential use in integrated pest control strategies.

**Table 4.** Efficacy of Natural Zeolites on Stored—Product Pests.

| Tested Crop | Type of the NZ | Affecting Insects | Reference |
|---|---|---|---|
| Rice | | *Oryzaephilus mercator* | Eroglu et al. [191] |
| Wheat | Minazel plus | *Rhyzopertha dominica* *Sitophilus oryzae* *Tribolium castaneum* | Kljajic et al. [192] |
| Maize | | *Sitophilus zeamais* *Sitophilus oryzae* | Haryadi et al. [181] |
| Chickpea | | *Lasioderma serricorne* | Perez et al. [193] |
| Oilseed (Rapeseed) | Klinofeed (dust) | *Meligethes* sp. | Daniel et al. [182] |

*3.9. Mycotoxin Control*

The use of aluminosilicates such as zeolites has emerged as a mycotoxin—binding agent in the feed and food industry to effectively adsorb mycotoxin [194]. Clinoptilolite has the capacity to adsorb aflatoxins by chelating of the β—dicarbonylmoiety in aflatoxin with uncoordinated metal ions [195]. There are some well—established criteria to evaluate the function of any binding additive, such as low inclusion rate, stability over a wide range of pH, huge capacity, and affinity to absorb various concentrations of mycotoxins [194]. The supplementation of mycotoxin binders in contaminated foods has been suggested as the most advantageous dietary approach to lower the mycotoxins efficacy [196]. Hydrated sodium calcium alumino—silicates—zeolite powder (HSCAS) has been identified as "aflatoxin—selective clay", but it does not adsorb other mycotoxins such as cyclopiazonic acid which may coexist with aflatoxin [197] while responses seem to be dose—dependent [198]. Parlat et al. [199] observed that clinoptilolite could successfully minimize the effects of aflatoxin in quail. Natural zeolites with high clinoptilolite content (over 80%) effectively adsorbed aflatoxin B1, aflatoxin B2, and aflatoxin G2 [200]. On the contrary, surface modified zeolites with $NH_4^+$ showed very well adsorption of ochratoxin A, T—2 toxin, zearalenone and aflatoxin B1 [201]. According to Adamovic et al. [202], the application of zeolites at 2 g kg$^{-1}$ of silage accelerates the fermentation and reduction of T—2 toxin, mould and zearalenone. Zeolites application as mycotoxin binder is impressive against aflatoxicosis, however, their effectiveness against trichothecenes, zearalenone and ochratoxin is restricted. At the same time, these compounds show high inclusion rates for vitamins and minerals, which are considered as one of the major disadvantages [203].

**4. Limitation of Zeolites**

Rather than the huge applicability of zeolites in agriculture, it should be considered that the zeolites are not without disadvantages. The fine—grained synthetic zeolites are highly dispersive in nature which creates worrisome problems during their use. After mining the usable form of natural zeolites is obtained via isolation procedures like crushing and pellet generation while the application of the synthetic form of zeolites are limited into hard, wear—resistant granular forms. The practical use of granular zeolites is not yet discovered [204]. The distribution of the zeolites sources is very limited such asthezeolitic soil is confined to only 1% of the total geographic area of India and more than 50% of natural zeolites are produced in China among all over the world [40] that may increase the price and the gap between demand and supply. Therefore, the uninterrupted availability of zeolites for farming purposes in worldwide is another major constraint.

**5. Future Scope**

The significant application of zeolites in agricultural activities has been well established by various researchers. However, systematic and comprehensive efforts are further needed for future research, including (a) precision mapping of the available zeolite deposits in each country, (b) determination of the physical stability of zeolites in various agro-climatic conditions, (c) economically viable organo-zeolitic manure or fertilizer devel-

opment, (d) evaluation of the risk of leaching of a toxic surfactant that is loosely attached to the zeolite surface, (e) assessment of the long-term impact of zeolite application on rhizospheric microflora and fauna, (f) understanding of the mechanisms of zeolite-mediated heavy metal stabilisation in contaminated soil and (h) development of zeolite-rich herbicides to minimise the residual risk hazard.

## 6. Conclusions

In the situation of rapid urbanisation and over-increasing population where resources are limited, there is no choice for us but to depend on agricultural productivity. In this context, various researchers suggest that farming with zeolites may be an option to improve soil's physical environments in terms of decreasing bulk density, increasing total porosity and increasing water-holding capacity. Furthermore, the existence of open networks in the zeolite structure leads to the formation of new routes for water movement, subsequently improving the infiltration rate and saturated hydraulic conductivity. Zeolites also show a strong affinity to various essential nutrient ions by modifying their surface chemistries using cationic surfactants, multifunctional adsorbents that have the capacity to trap anions and non-polar organics. Thus, the application of zeolite-loaded fertilizer improves the nutrient retention in soil and releases nutrients slowly throughout the crop life; otherwise, rapid mineralisation would take place, leading to nutrient loss. Zeolites are very much effective in remediation of heavy metal toxicity and wastewater treatment, and they could help to improve soil's biological properties. Zeolite application in space missions as zeoponic substrates opens a new dimension of zeolites. The aforesaid positive impacts ultimately enhance crop growth, productivity and even quality attributes of various agronomic and horticultural crops. The higher input use efficiency significantly reduces greenhouse gas emissions and energy involvement and facilitates better carbon sequestration. However, the impact of zeolite application varies with the agro-climatic location, the nature of zeolites, their availability and application strategies, and soil textural classes. Further studies are needed to identify zeolite resources and the long-term impact on the soil environment and to develop new, cost-effective zeolite-based nutrient resources for sound agricultural practices.

**Author Contributions:** Conceptualization, M.M.; B.B.; and S.G.; writing—original draft preparation, M.M.; S.G.; S.S.; H.B.; and A.H.; writing—review and editing, B.B.; S.M.; K.B.; P.K.B.; H.B., A.H., M.S., P.O.; and M.B.; funding acquisition, P.O., M.S., A.H. and M.B. All authors have read and agreed to the published version of the manuscript.

**Funding:** This research was funded by the 'Slovak University of Agriculture', Nitra, Tr. A. Hlinku 2,949 01 Nitra, Slovak Republic under the project 'APVV—18—0465 and EPPN2020—OPVaI—VA—ITMS313011T813'.

**Conflicts of Interest:** The authors declare no conflict of interest.

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
