# Peer review of "Zeolites Enhance Soil Health, Crop Productivity and Environmental Safety"

_agronomy, doi:10.3390/agronomy11030448_

Round 1
Reviewer 1 Report
The manuscript "Zeolites Enhance Soil Health, Crop Productivity and Environmental Safety" is a well-written review. Comprehensive information has been collected and organized in this paper. I believe this manuscript has reported the current advances for zeolite application in many areas. The drawback of this paper is that during the reading I felt it seems like a manual or memo about who had used the zeolite and what the results. Theoretical illustration and the thoughts from the authors were largely lacking.
I have marked the errors and some comments in the attached file.

Author Response
Response to Reviewer 1 comments
Comments: Need to check the price for the word ‘Inexpensive’ |
Authors’ response: We agree with the comment and remove this word |
Comments: No need to show Figure 2 |
Authors’ response: With due respect, we are not agreed with this comment as the figure clearly shows the increasing trends of zeolite application over the years |
Comments: No need to show Figure 3 |
Authors’ response: With due respect, we think Fig. 3 indicates the limited availability of zeolite which is a serious concern. |
Comments: Copyright issue in figure 4 |
Authors’ response: No it is the modified form and we have added the proper source |
Comments: Full term for CEC? |
Authors’ response: We have added the full form of CEC |
Comments: What is cmol? |
Authors’ response: It is centimol which is the unit of CEC. |
Comments: redundant with the context in line 84-86. |
Authors’ response: Thanks for this comment. We have corrected this line as par comment |
Comments: What does dSm-1 unit mean? |
Authors’ response: It means the deci Siemens per metre which is the unit of EC that indicates the presence of salt. It is the standard and International Unit (SI) for the EC. We have mentioned it in manuscript. |
Comments: How can figure 7 be modified? |
Authors’ response: We made this graphs by taking some of the data (significant findings) from the manuscript of the Razmi and Sepaskhah [61]. We did not use any actual figure on previously published content in original from. |
Comments: The table 2 style needs to be modified according to the standards of the journal. |
Authors’ response: Thanks for this comment and we have formatted the table according to journal standard |
Comments: The point of 3.1 should be 3.2 |
Authors’ response: Thanks for this comment and we have corrected as par comment |
Comments: Unit correction in the line number 263 |
Authors’ response: We have corrected the unit |
Comments: Full form of GHGs |
Authors’ response: We put the full form of GHGs |
Comments: Awkward sentence. Please rephrase the lines between 291-293 |
Authors’ response: We have rephrased the sentence as suggested. |
Comments: Please explain more how the organic acids absorption and odor releasing affect ammonium loss. |
Authors’ response: Thanks for this comment. The actual reasons behind the low ammonia losses due to the use of zeolite during composting are clearly described in the line between 305-312. There was a mistake of sentence writing and we have corrected it. |
Comments: How this process affects plants taking in ammonia? |
Authors’ response: When the loss of N as well as P from soil is reduced by zeolitic amendments, the nutrients availability will prolonged and plant can effectively uptake and utilize throughout its growing period. |
Comments: This reaction needs to be explained more. |
Authors’ response: We have explained the reaction in line 333-334 |
Comments: The o4 should be O4. |
Authors’ response: Thanks for this comment and we have corrected the figure |
Comments: What is this for the line 370? |
Authors’ response: The two data are separate for Zn and Mn and it has been mentioned |
Comments: What is DTPA? |
Authors’ response: DTPA is Diethylenetriamine pentaacetate. Properly explained in line 370-371 |
Comments: What are the meanings of a, b, and ab in figure 9 |
Authors’ response: The letters indicate the DMRT values and we have clearly mentioned it. |
Comments: No necessary to show the unpublished data as a figure 10. |
Authors’ response: We had written mistakenly that the figure was drawn from unpublished data but actually, it is the modified form of figure and we have also mentioned the proper source in figure title. |
Comments: Please note the full term of SOM |
Authors’ response: We have mentioned the full form of SOM in line 399 |
Comments: The point 3.3 should be 3.4. |
Authors’ response: Thanks for this comment and we have corrected it. |
Comments: The point 3.4 should be 3.5. |
Authors’ response: Thanks for this comment and we have corrected it. |
Comments: Why the word dissolute was used? |
Authors’ response: The word was written by mistake and we have corrected in revised version |
Comments: Reviewer asked why the limited distribution of zeolitic soil is another constraint? |
Authors’ response: The source of natural zeolite is mainly confined into the some parts of the world and not evenly distribution which we have already mentioned in line 108-109. This limitation may increase the price of zeolite and reduce the availability |
Comments: Reviewer does not agree with the point that ‘additionally, it has been well reported that natural zeolites mainly deposit in the semi-arid region, making restricted availability for other climatic zones’. |
Authors’ response: Thanks for this comment and we have agreed with this and removed the lines from manuscript |
Comments: Why the precision mapping of the available zeolites deposit in each country is important for zeolite application? |
Authors’ response: As the sources of natural zeolite are very limited so it is necessary to find out the deposition of zeolite with the help of précised advanced technology. |

Reviewer 2 Report
The article presented as Review, some application of zeolites in agriculture, These application are well documented and are presented in Chapters as Origin; Impact of zeolite application in agriculture (i.e. Improvement of soil physical properties; Nutrients retention; Environmental impact; Slow release of fertilizers; Remediation of contaminated soil; Wastewater treatment; Crop management practice; Used as pesticides; Mycotoxins Control) Limitation of zeolites; Future scope.
The article is interesting and can be useful for a specialist from Agronomy...but, before publishing, need some revisions:
A) Figure 1, Figure 2, Figure 3, Figure 4 appear to be copied from another source -(possible plagiarism) that because there does not appear the phrase: ''Modified from'' . If that is true, then the pictures must be presented in another way. Or if they have the approval from original source, then they must write '' with approval of...).
B) In Table 1: the bibliographic sources must be given in the table, eventually in another column ( in the same way as Table 2);
C) Chapter 3.1. appear twice...so on page 9 the number 3.1 must be replaced with 3.2;
D) the text is not the same format in the manuscript (the text is not aligned in the same way);
E) at row 291-293: the phrase must be revised;
F) Chemical reaction give on page 11 must be numbered;
G) the scientific name of plants and insect must be written with italics characters;
H) At the row. 630 the phrase ''zeolite that belongs from...'' must be replaced with '' zeolite which contains''
Author Response
Response to Reviewer 2 comments
Comments: Figure 1, Figure 2, Figure 3, Figure 4 appear to be copied from another source -(possible plagiarism) that because there does not appear the phrase: ''Modified from'' . If that is true, then the pictures must be presented in another way. Or if they have the approval from original source, then they must write '' with approval of...). |
Authors’ response: Thanks for the comments. These figures are not copied from other published sources. We made these figures by taking some of the data (significant findings) from the manuscript of the respective authors and also cited them accordingly. We did not use any actual figure(s)/copied content on previously published content as the original form. All figures are solely made by the authors. |
Comments: In Table 1: the bibliographic sources must be given in the table, eventually in another column ( in the same way as Table 2) |
Authors’ response: Many thanks for the comments. We have added another row for references |
Comments: Chapter 3.1. appear twice...so on page 9 the number 3.1 must be replaced with 3.2; |
Authors’ response: Thank you for the comment. We have corrected the chapter number |
Comments: The text is not the same format in the manuscript (the text is not aligned in the same way); |
Authors’ response: We have tried to maintain the journal standard format and alignment throughout the manuscript. |
Comments: at row 291-293: the phrase must be revised |
Authors’ response: Thanks for this comment and we have revised the sentence. |
Comments: Chemical reaction give on page 11 must be numbered |
Authors’ response: We have added the number in reaction |
Comments: The scientific name of plants and insect must be written with italics characters |
Authors’ response: Thanks for this comment and we have written the scientific names of plants and insects with italics characters |
Comments: At the row. 630 the phrase ''zeolite that belongs from...'' must be replaced with '' zeolite which contains'' |
Authors’ response: Many thanks for this comment. We have corrected the sentence as suggested. |

Reviewer 3 Report
Figure 1 - No data source
Figure 3 - Are the names in the picture regions in India?
Line 110 - Regions or provinces in India - please describe
Line 257 of the chapter number should be 3.2 - no 3.1
Line 411, chapter number 3.3 or 3.4 And probably the consecutive numbering of the chapters should be changed
Lines 412-421 Chapter - "Slow Release of Herbicides" - There is no clear explanation of what the effect of zeolites on herbicides is - what effect this has on weed control with these actives - whether the herbicidal effect is prolonged or, on the contrary, whether the effectiveness of weed control is reduced?
In the "Crop Management Practices" chapter, it would be very valuable to develop a drawing or table showing the effect of zeolites on the yield of the main crops mentioned in this chapter.
Verse 631-632 It would be good to give the Latin names of the pests mentioned in the text
In Table 4, please include the full Latin names of the pests
Author Response
Response to Reviewer 3 comments
Comments: Figure 1 - No data source |
Authors’ response: Figure 1 represents the overall advantages of zeolite application. This figure was exclusively made by the authors based on the knowledge and from various literate/ research papers. |
Comments: Figure 3-Are the names in the picture regions in India? |
Authors’ response: Yes the names in the picture are the states in India |
Comments: Line 110 - Regions or provinces in India - please describe |
Authors’ response: We have mentioned in line 117 |
Comments: Line 257 of the chapter number should be 3.2 - no 3.1 |
Authors’ response: Thanks for this comment and we have corrected it |
Comments: Line 411, chapter number 3.3 or 3.4 And probably the consecutive numbering of the chapters should be changed |
Authors’ response: Thanks for this comment and we have corrected it |
Comments: Lines 412-421 Chapter - "Slow Release of Herbicides" - There is no clear explanation of what the effect of zeolites on herbicides is - what effect this has on weed control with these actives - whether the herbicidal effect is prolonged or, on the contrary, whether the effectiveness of weed control is reduced? |
Authors’ response: Many thanks for this comment and we have tried to include a clear explanation regarding the effect of zeolite on herbicide, its roles and its importance. |
Comments: In the "Crop Management Practices" chapter, it would be very valuable to develop a drawing or table showing the effect of zeolites on the yield of the main crops mentioned in this chapter. |
Authors’ response: We have already included the figures (11-16) that indicates the yield, yield attributing traits and quality characteristics of major crops. |
Comments: Lines 412-421 Chapter - "Slow Release of Herbicides" - There is no clear explanation of what the effect of zeolites on herbicides is - what effect this has on weed control with these actives - whether the herbicidal effect is prolonged or, on the contrary, whether the effectiveness of weed control is reduced? |
Authors’ response: Proper explanation was added accordingly. The slow-release nature of herbicide when used with zeolites improves the herbicide efficiency to control the weed floras and the prolonged effect of herbicide keeps the weed-free crop field throughout the entire crop weed competition period. Zeolite rich nanocapsule is used as a herbicide carrier, adsorbent and retaining agent. A longer retention period of zeolite added herbicide on weed leaves helps in maximizing the efficacy of the herbicidal mode of action. |
Comments: Verse 631-632 It would be good to give the Latin names of the pests mentioned in the text |
Authors’ response: Thanks for this comment and we have included the Latin names |
Comments: In Table 4, please include the full Latin names of the pests |
Authors’ response: We have added the full Latin names of the pests in Table 4 |
